# Cancer treatment monitoring using cell-free DNA fragmentomes

Iris van 't Erve [1,7], Bahar Alipanahi[2,7], Keith Lumbard[2,7], Zachary L. Skidmore [2,7], Lorenzo Rinaldi[2], Laurel K. Millberg [2], Jacob Carey[2], Bryan Chesnick[2], Stephen Cristiano[2], Carter Portwood[2], Tony Wu[2], Erica Peters[2], Karen Bolhuis[3], Cornelis J. A. Punt[4], Jennifer Tom[2], Peter B. Bach[2], Nicholas C. Dracopoli[2], Gerrit A. Meijer [1], Robert B. Scharpf[5], Victor E. Velculescu [5], Remond J. A. Fijneman [1] ✉ & Alessandro Leal [2,6] ✉

Circulating cell-free DNA (cfDNA) assays for monitoring individuals with cancer typically rely on prior identification of tumor-specific mutations. Here, we develop a tumor-independent and mutation-independent approach (DELFI-tumor fraction, DELFI-TF) using low-coverage whole genome sequencing to determine the cfDNA tumor fraction and validate the method in two independent cohorts of patients with colorectal or lung cancer. DELFI-TF scores strongly correlate with circulating tumor DNA levels (ctDNA) (r = 0.90, $p < 0.0001$, Pearson correlation) even in cases where mutations are undetectable. DELFI-TF scores prior to therapy initiation are associated with clinical response and are independent predictors of overall survival (HR = 9.84, 95% CI = 1.72-56.10, $p < 0.0001$). Patients with lower DELFI-TF scores during treatment have longer overall survival (62.8 vs 29.1 months, HR = 3.12, 95% CI 1.62-6.00, $p < 0.001$) and the approach predicts clinical outcomes more accurately than imaging. These results demonstrate the potential of using cfDNA fragmentomes to estimate tumor burden in cfDNA for treatment response monitoring and clinical outcome prediction.

The measurement of plasma circulating tumor DNA (ctDNA) from liquid biopsies has emerged as a minimally invasive biomarker for tumor detection and therapeutic monitoring[1–3]. ctDNA burden can change over the course of the disease, decreasing upon treatment response and increasing as the tumor develops resistance to therapy[3,4]. Plasma-derived ctDNA is a dynamic tumor marker due to its short half-life and may be able to detect relapse earlier than imaging and clinical parameters[5–7]. Monitoring ctDNA dynamics throughout treatment can enable physicians to make timely and informed treatment decisions[7,8]. Doing so, however, requires a rapid, inexpensive, and generally applicable monitoring test that predicts therapeutic success and clinical outcomes.

A variety of technologies exist for measuring ctDNA. Ultra-deep next-generation sequencing (NGS) of a targeted set of genes is an approach that can provide information about somatic abnormalities and detect a tumor's genomic changes. However, this method has limitations due to the confounding signal of clonal hematopoietic variants that arise in aging individuals[9]. Tissue-guided or white blood cell-informed approaches can be used to filter out these variants and prevent them from obscuring the detection of tumor-specific

[1]Department of Pathology, Netherlands Cancer Institute, Amsterdam, The Netherlands. [2]Delfi Diagnostics, Inc., Baltimore, MD, USA. [3]Department of Medical Oncology, Amsterdam UMC, Cancer Center Amsterdam, University of Amsterdam, Amsterdam, the Netherlands. [4]Julius Center for Health Sciences and Primary Care, University Medical Center Utrecht, Utrecht, The Netherlands. [5]The Sidney Kimmel Comprehensive Cancer Center, Johns Hopkins University School of Medicine, Baltimore, MD, USA. [6]NYU Langone Health Perlmutter Comprehensive Cancer Center, New York, NY, USA. [7]These authors contributed equally: Iris van 't Erve, Bahar Alipanahi, Keith Lumbard, Zachary L. Skidmore. ✉e-mail: r.fijneman@nki.nl; alessandro.leal@nyulangone.org

alterations[10], however, such approaches are logistically complex and costly. Evaluation of single nucleotide variants can be achieved via less expensive ctDNA hotspot mutation approaches, such as droplet digital PCR (ddPCR), but such technologies are limited as they can only evaluate one or a few variants per reaction, requires prior knowledge that a patient's tumor harbors a mutation of interest, and cannot be easily scaled in a patient population with heterogeneous tumor mutations[11–15].

It is well known that the genomic and chromatin characteristics of cancers are different from normal cells[16]. These observations have led to the development of an approach called DELFI (**D**NA **E**va**L**uation of **F**ragments for Early **I**nterception), which examines cfDNA to detect various cancers and indicate their tissue of origin[17]. Due to the widespread nature of the underlying changes leading to altered cfDNA fragmentomes in patients with cancer, we evaluated whether DELFI could be of added clinical value for monitoring disease progression. In this study, we developed DELFI-tumor fraction (DELFI-TF), a machine-learning-based approach capable of non-invasively determining tumor burden without requiring a priori genetic information from the tumor. We demonstrate the feasibility of this method and evaluate its utility for monitoring disease and treatment responses in patients with metastatic colorectal cancer (mCRC) and lung cancer.

## Results

### DELFI-TF model development using genome-wide cfDNA fragmentation profiles

To develop a mutation- and tumor-independent approach for quantifying circulating tumor burden using genome-wide cfDNA fragmentation data, we analyzed 689 longitudinal plasma samples (pre- and on-treatment) from 153 patients with unresectable liver-limited mCRC in the prospective phase III CAIRO5 clinical trial[18] (Supplementary Data 1 and Supplementary Figs. 1, 2). Patients were treated with a fluoropyrimidine-based first-line regimen (FOLFOX or FOLFIRI) plus bevacizumab (Fig. 1a, Supplementary Fig. 1, and Supplementary Data 1). As part of the trial, the mutation status of *KRAS* (exon 2, 3, and 4), *NRAS* (exon 2 and 3), and *BRAF* (codon 600) was previously assessed in tumor tissue. In the current study, we assessed 79 patients with *KRAS/NRAS/BRAF* mutant tumors and 74 patients with *KRAS/NRAS/BRAF* wild-type tumors. We assessed the tumor tissue-informed mutation status in the pre-treatment and longitudinal cfDNA samples for patients with *RAS/BRAF* mutant tumors. In total, tumor-informed mutational analyses of cfDNA were performed in 309 samples from patients with *RAS/BRAF* mutant tumors, while DELFI-TF analyses were performed in 689 samples from patients in both *RAS/BRAF* mutant and wild-type tumors (Supplementary Fig. 2). DELFI-TF failure rates associated with library preparation and whole-genome sequencing (WGS) were <1% (Supplementary Fig. 2).

An analysis of fragmentation features and arm-level chromosomal changes in patients with mCRC revealed dramatic alterations in these features across the genome for the vast majority of patients at baseline as well as at time points associated with progressive disease, regardless of demographic or clinical characteristics (Fig. 1b, c). In contrast, the majority of patient samples at time points associated with stable disease or radiologic response after the start of first-line systemic therapy were associated with fewer fragmentome or genomic abnormalities (Fig. 1b, c). These findings suggested that the fragmentome-based model may be capable of real-time identification of systemic treatment response in a non-invasive manner.

To examine the origins of cfDNA fragmentation patterns in this study, we compared genome-wide fragmentome profiles with chromosome conformation capture (Hi-C) open (A) and closed (B) compartments for colorectal cancer cells as well as normal blood cells. We found that cfDNA patterns of healthy individuals (Track 4) were highly correlated to those of lymphoblastoid cells (Track 5). Analysis of short/long ratio profiles from pre-treatment samples of 10 CRC patients with high DELFI-TF scores (Tracks 2 and 3) revealed that their fragmentome had high similarity to A/B compartments previously estimated from CRC tissue reference samples (Track 1). Notably, this pattern is observed in regions of the genome that are largely copy-neutral in these samples (green bars). These analyses suggested that cfDNA fragmentomes from individuals with CRC represent a mixture of cfDNA profiles of chromatin compartments of cells from peripheral blood as well as those from colorectal cancer (Supplementary Fig. 3).

We evaluated both mutation levels and genome-wide fragmentation features from the same samples in this large cohort to develop a fragmentomics-based approach for quantifying circulating tumor fraction (Fig. 2a). For all plasma samples (n = 309) from patients who had *RAS/BRAF*-mutant tumors, the cfDNA tumor burden was determined using the mutant allele frequency (MAF) of the tumor-informed *RAS/BRAF* mutation as measured by ddPCR (Supplementary Data 2). Using a second aliquot of the same cfDNA sample, low-coverage whole-genome sequencing (~6x) was performed to obtain fragmentome characteristics (Fig. 2a and Supplementary Data 2). Random forest regression models (RFs) were trained and cross-validated to predict tumor-specific MAFs from longitudinal cfDNA samples with *RAS/BRAF* mutations. Features of the RFs included characteristics of genome-wide fragmentation[17], chromosomal arm changes[17,19], and mixture component weights for the overall cfDNA fragment sizes (Fig. 2a and Supplementary Data 3). All these features contributed to the DELFI-TF machine-learning algorithm, with chromosomal changes comprising ~55%, mixture model ~32%, and short to long (S/L) features ~13%. We then trained an RF model on all samples with *RAS/BRAF* mutations and applied this model to cfDNA samples of patients with *RAS/BRAF* wild-type tumors.

### DELFI-TF accurately reflects ctDNA mutant allele frequencies and copy number changes

We assessed DELFI-TF scores from cancer patients in the CAIRO5 cohort as well as from individuals without cancer who were part of a previously reported screening cohort in Denmark[20] (n = 153) (Supplementary Data 4). Baseline samples (n = 128) from the CAIRO5 cohort had a median DELFI-TF score of 25% (95% confidence interval (CI) = 21–30%), while samples from individuals without cancer (n = 153) exhibited DELFI-TF scores close to zero (non-cancer DELFI-TF = 0.09% (95% CI = 0.08–0.13%), p < 0.0001 compared to CAIRO5 baseline samples, Wilcoxon rank-sum). Notably, all of the baseline samples from patients with mCRC had DELFI-TF scores higher than 0.13%, the upper 95% CI bound of individuals without cancer (Fig. 2b). At the first time point after the initiation of therapy (T1, n = 151), DELFI-TF levels were significantly reduced compared to baseline samples, but still significantly higher than those without cancer (T1 DELFI-TF = 0.45% (95% CI = 0.30–0.63%), p < 0.0001 compared to non-cancer, Wilcoxon rank-sum) (Fig. 2b). To independently validate the DELFI-TF approach, we used the locked model to analyze genome-wide sequence profiles of all plasma samples from patients with *RAS/BRAF* wild-type tumors in the CAIRO5 cohort and observed similar score distributions among the clinical groups assessed (Fig. 2b).

Next, we assessed the analytical performance of DELFI-TF for measuring tumor burden in patients with *RAS/BRAF* mutant tumors using ddPCR for MAF quantification. A strong correlation was observed between DELFI-TF and ddPCR across a range of positive MAF values (r = 0.90, p < 0.0001, Pearson correlation) (Fig. 2c). Surprisingly, we were able to observe DELFI-TF scores which had undetectable ddPCR measurements for *RAS/BRAF* alterations for both baseline (n = 5) as well as on-treatment samples (n = 127). To provide evidence that DELFI-TF can detect ctDNA even when mutations were undetectable, we analyzed the samples (n = 45) with undetectable mutations with the ichorCNA approach[21] and evaluated those cases with ichorCNA scores >5%, as this method would be expected to be accurate above this threshold (Supplementary Fig. 4). We observed a high

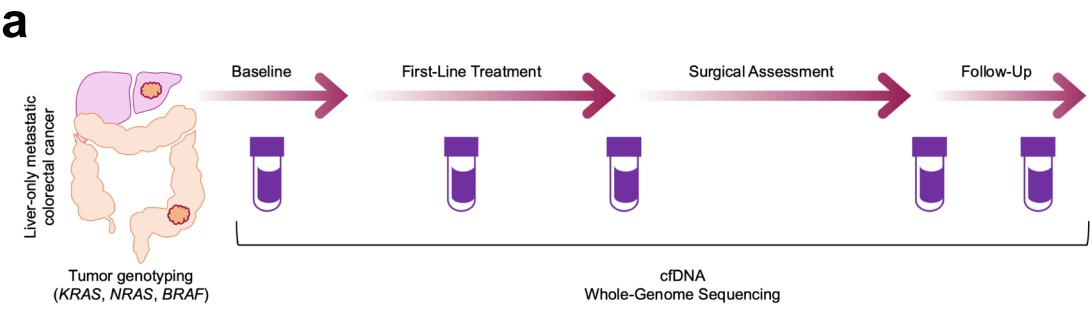

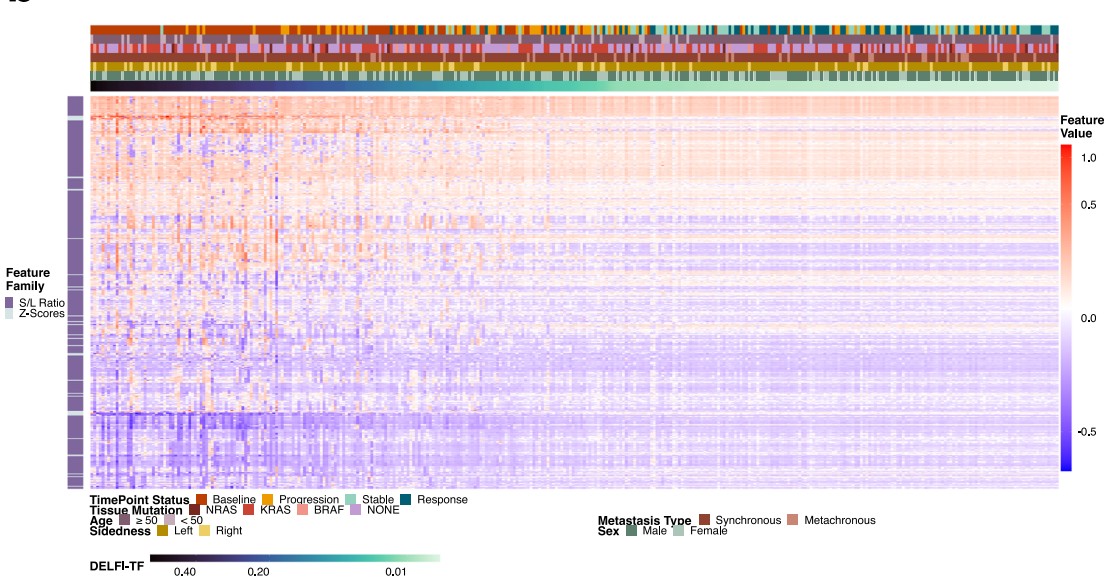

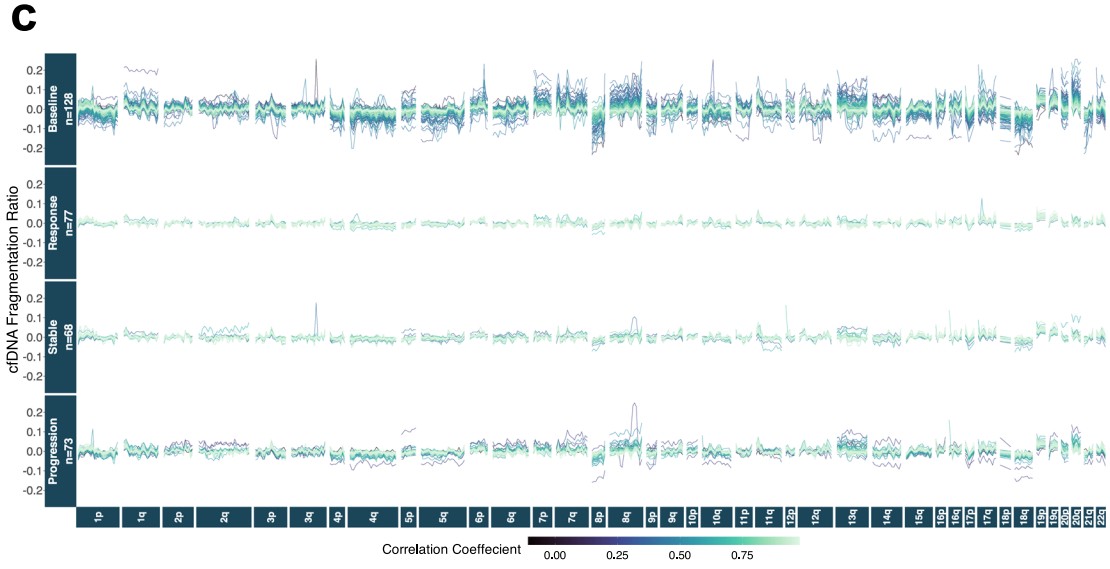

**Fig. 1 | DELFI-tumor fraction (DELFI-TF) as a mutation-independent approach for tumor monitoring. a** Tumors from patients with treatment-naive non-operable liver-only mCRC who enrolled in the CAIRO5 phase III trial were tested for hotspot mutations in KRAS, NRAS, and BRAF. Blood samples were collected at baseline, during treatment, and at the time of disease progression or last follow-up. Patients carrying *KRAS, NRAS,* or *BRAF* driver mutations were monitored with ddPCR and DELFI-TF assays. Patients with wild-type *KRAS, NRAS,* and *BRAF* tumors were monitored with DELFI-TF only. **b** Heatmap representation of genomic features depicts deviations of cfDNA fragment ratios and chromosomal arm-level z-scores across baseline and on-treatment time points of 149 patients having a liquid biopsy within 60 days of a RECIST1.1 evaluation, along with DELFI-TF values and clinical and demographic characteristics. **c** cfDNA genome-wide fragmentation profiles in 504 non-overlapping 5 Mb genomic regions at baseline and at time points within 60 days of imaging assessment by RECIST1.1 show marked heterogeneity at baseline and for patients who exhibited disease progression compared to patients who experienced stable disease or radiologic response after initiating first-line systemic therapy. Correlations for fragment ratios across 504 genomic regions for each sample compared to identical genomic regions in 153 non-cancers are shown in the color scale.

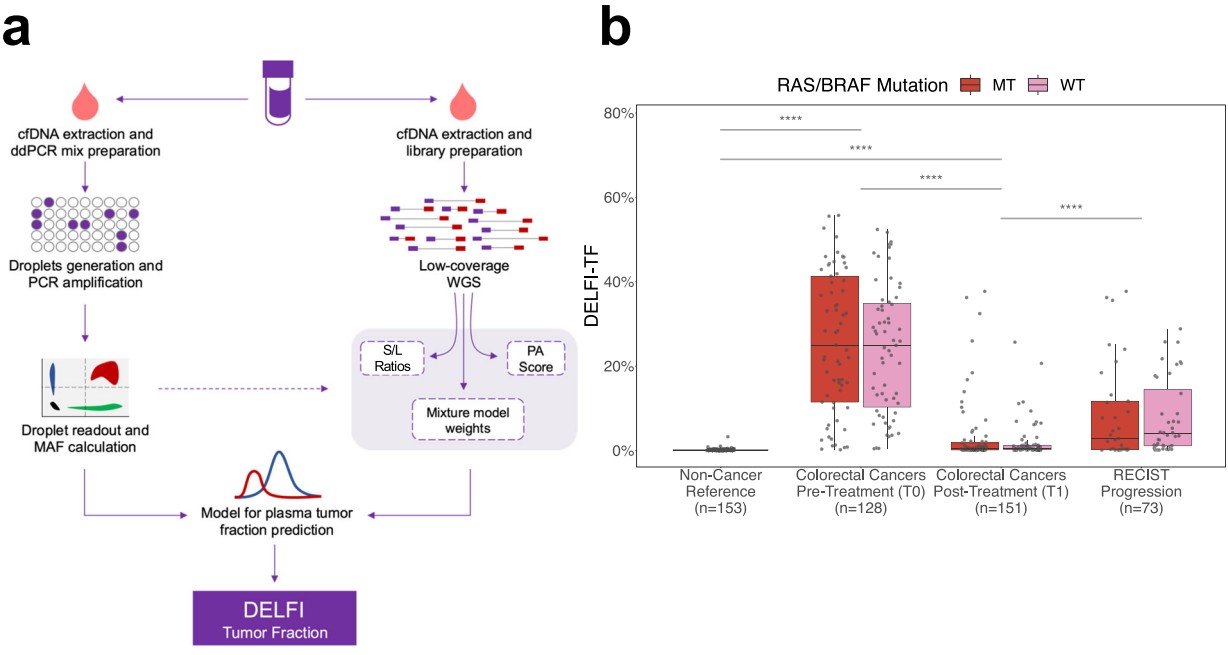

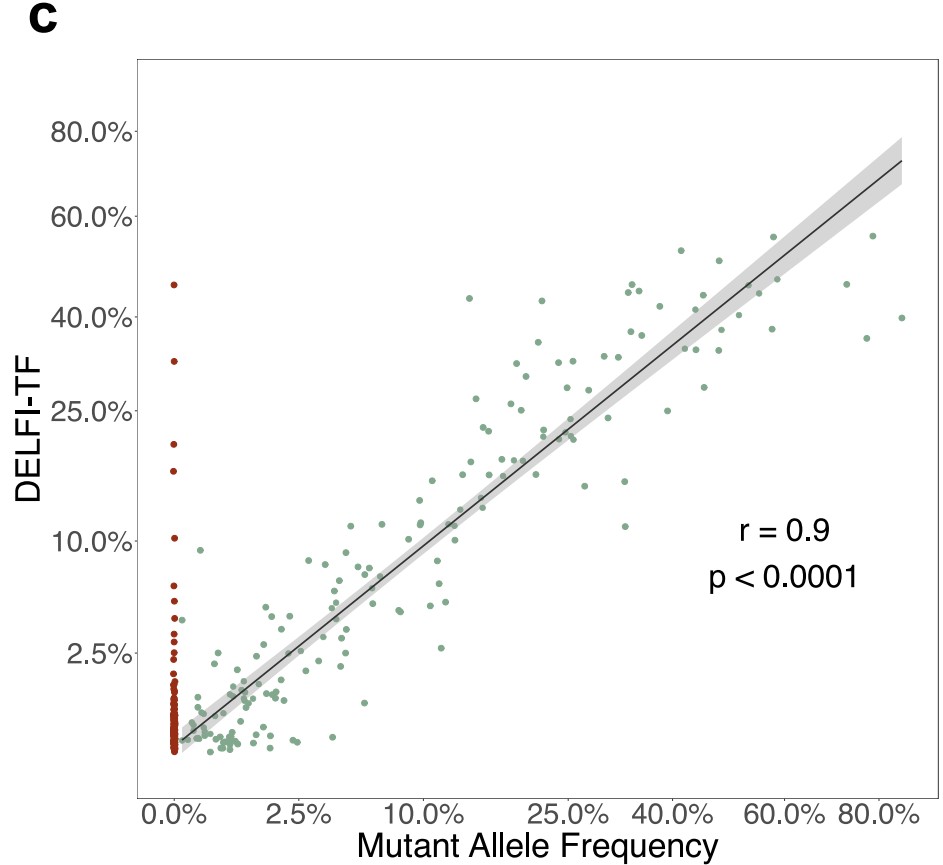

correlation ($r = 0.95$, $p < 0.0001$, Pearson correlation) of DELFI-TF and ichorCNA values in these samples, strongly supporting the notion that these patients did indeed have detectable ctDNA that was missed using ddPCR (Supplementary Fig. 4).

To determine whether DELFI-TF provided an advantage over existing methods to evaluate cfDNA, we compared our approach to the ichorCNA[21] WGS method for all samples with detectable MAF levels by

ddPCR. We compared ichorCNA, DELFI-TF, and MAF for all applicable samples in the CAIRO5 study MT arm across four quartiles of MAF values. We observed high concordance for DELFI-TF and MAF with ichorCNA at high MAF levels (quartiles 3 and 4, >4.8%) (DELFI-TF vs. ichorCNA, $r = 0.77$, $p < 0.0001$; MAF vs. ichorCNA, $r = 0.51$, $p < 0.0001$, Pearson correlation) (Supplementary Fig. 4). However, at lower MAF levels (quartiles 1 and 2, ≤4.8%), while DELFI-TF and MAF values were

**Fig. 2 | DELFI-TF accurately predicted cfDNA tumor burden in patients with metastatic colorectal cancer. a** Model schematic. Plasma aliquots were collected from patients and used for cfDNA isolation. From each time point of patients with tissue-confirmed RAS/BRAF mutations, duplicate cfDNA samples were utilized for ddPCR and low-coverage WGS. WGS fragment-sequencing statistics were calculated per sample at a given time point. A random forest model was trained against the MAFs called by ddPCR readouts of the tumor-specific *RAS/BRAF* variants in all longitudinal cfDNA samples to generate the DELFI-TF values. **b** Patients with mCRC exhibit a wide range of DELFI-TF values at baseline (T0; *n* = 128) and reduced tumor fractions at the first time point after treatment commencement (T1; *n* = 151). Tumor fractions increase from treatment commencement (T1) to progression (*n* = 73). Non-cancer controls exhibit remarkably low DELFI-TF values (*n* = 153). No

significant difference was observed between mutant type (MT) and wild-type (WT) samples for either time point (T0, *p* = 6.10e-01, T1, *p* = 5.14e-01, progression, *p* = 4.57e-01, two-sided Wilcoxon rank-sum). **c** DELFI-TF strongly correlates with detectable MAF values measured by ddPCR (*n* = 177, *r* = 0.90, *p* = 8.83e-65, two-sided Pearson correlation). Plasma time points with undetectable MAF (*n* = 132; *n* = 5 at T0, 40 at T1, 28 at T2, 59 at T3–T9) are indicated in red and included those with high DELFI-TF values in patients which were independently validated as having measurable tumor burden. The middle hinge in the boxplots corresponds to the median, while the lower and upper hinges correspond to the first and third quartiles. The upper whisker extends from the hinge to the largest value no further than the 1.5 * interquartile range from the hinge. Ribbons around the regression line in correlation plots represent the 95% confidence level interval for predictions.

still significantly correlated (*r* = 0.60, *p* < 0.0001, Pearson correlation), ichorCNA and MAF values were not correlated (*r* = 0.02, *p* = 0.87, Pearson correlation), and ichorCNA values exhibited a plateau at lower MAF values of ~5%. A similar relationship between DELFI-TF and ichorCNA MAF predictions were observed in the WT arm (Supplementary Fig. 4).

We examined the association between cfDNA fragmentomes and copy number changes in tumor tissue and plasma samples of the same patients (Supplementary Fig. 5). By analyzing the copy number profile of the tissue samples, we observed abnormal cfDNA fragmentation profiles in regions of the genome that were copy-neutral in tumor tissues and were further altered in regions with copy-number changes (Supplementary Fig. 5). Analysis of two genes with common copy number alterations in mCRC (*MBD1* and *PLGC1*) in plasma revealed high correlations between DELFI-TF or ddPCR tumor fraction and the copy number ratio at these regions in cfDNA (*n* = 79, Supplementary Data 5). Importantly, DELFI-TF was negatively correlated with *MBD1* deletions (DELFI-TF, *r* = −0.79, *p* < 0.0001; ddPCR MAF, *r* = −0.67, *p* < 0.0001, Pearson correlation), and positively correlated with *PLGC1* amplifications (DELFI-TF *r* = 0.59, *p* < 0.0001; ddPCR MAF *r* = 0.55, *p* < 0.0001, Pearson correlation) (Supplementary Fig. 5). Altogether, these analyses demonstrate that DELFI-TF accurately captures ctDNA fraction as assessed through detection of cancer-specific mutations or copy number aberrations in cfDNA.

To validate the analytical performance of the DELFI-TF model in an independent cancer cohort, we examined longitudinal time points from a series of patients with stage III or IV non-small cell lung cancer receiving first-line treatment with chemotherapy alone or in combination with immune checkpoint inhibition (Supplementary Fig. 5). A total of 47 cfDNA samples from 15 patients were analyzed using genome-wide DELFI-TF analyses as well as the validated elio plasma complete 2.2 Mb panel of common cancer driver genes with deep targeted sequencing (~25,000x coverage). We observed that the DELFI-TF scores at baseline and on-treatment time points strongly correlated with the highest MAF values (max MAFs) detected in the targeted sequencing data (*r* = 0.93, *p* < 0.0001, Pearson correlation) (Supplementary Fig. 5 and Supplementary Data 6).

### DELFI-TF at baseline and association with clinical characteristics and outcomes
We compared the molecular measurements of tumor burden obtained with DELFI-TF with the observed patient clinical characteristics in the CAIRO5 cohort. At the baseline time points, DELFI-TF and ddPCR MAF values were correlated with the sum of the longest diameters (SLD) of the target lesions in the liver (DELFI-TF *r* = 0.4, *p* < 0.001; ddPCR MAF *r* = 0.34, *p* < 0.01; Pearson correlation) (Supplementary Fig. 6 and Supplementary Data 2, 7). In contrast, no significant correlation was seen between these markers and serum carcinoembryonic antigen (CEA) levels measured at this time point (Supplementary Fig. 6).

Baseline DELFI-TF and ddPCR MAF values predicted subsequent clinical response as assessed by imaging, with patients with complete

or partial response (CR/PR) exhibiting substantially lower DELFI-TF or MAF scores than those with progression of disease or stable disease (PD/SD) (median DELFI-TF for CR/PR = 11.43% and for PD/SD = 30.26%, *p* < 0.05; median MAF for CR/PR = 8.29% and for PD/SD = 25.46%, *p* < 0.05, Wilcoxon rank-sum) (Fig. 3a). Similarly, patients with resectable liver lesions had significantly lower baseline tumor fractions as assessed by DELFI-TF and ddPCR (*p* < 0.05 for both, Kruskal–Wallis) (Fig. 3b) and DELFI-TF was lower in patients with metachronous compared to synchronous metastatic disease (*p* < 0.001, Wilcoxon rank-sum) (Fig. 3c). DELFI-TF and ddPCR tumor fractions were lower in patients who never had disease progression at any time point during the CAIRO5 trial (never-progressors) than in patients who experienced progressive disease at some point during treatment (ever-progressors), while imaging analyses could not distinguish between these groups (Supplementary Fig. 7).

At baseline, patients with DELFI-TF values in the first quartile showed longer median overall survival (OS) and progression-free survival (PFS) than patients with DELFI-TF above the first quartile (OS not reached vs 19 months, hazard ratio (HR) = 3.26, 95% CI = 1.46–7.29, *p* < 0.01, log-rank; PFS 13.38 vs 8.28, HR = 2.27, 95% CI = 1.22–4.26, *p* < 0.01, log-rank) (Fig. 3d and Supplementary Fig. 7). The OS and PFS survival curves for the MAF analyses were very similar to those with DELFI-TF, with baseline MAF values for tumor fraction assessment showing similar distinction in median OS (35.7 vs 19.0 months, HR = 2.82, 95% CI = 1.31-6.07, *p* < 0.01, log-rank) and median PFS (14.37 vs 8.28 months, HR = 2.82, 95% CI 1.46–5.45, *p* < 0.01, log-rank) (Fig. 3d and Supplementary Fig. 7). In contrast, serum CEA levels at baseline were unable to predict disease progression or death (Supplementary Fig. 7). A multivariate analysis model revealed DELFI-TF scores at baseline as a significant predictor of outcome, with higher HR compared to sidedness, age, CEA levels, or tumor size by SLD analyses (HR = 9.84, 95% CI = 1.72–56.1, *p* < 0.0001, log-rank) (Table 1).

### DELFI-TF dynamic changes during therapy and patient outcomes
Based on analyses of DELFI-TF prior to treatment initiation, we explored whether dynamic changes of this biomarker would reflect treatment response during therapy. We examined ever-progressors during therapy and observed that these individuals more often exhibited increasing DELFI-TF values at early time points and emerging disease progression at late time points than never-progressors (*p* < 0.05, Wilcoxon rank-sum) (Supplementary Fig. 8). A temporal analysis of DELFI-TF and ddPCR MAFs revealed comparable tumor dynamics, even at late time points in patients treated with curative-intent liver metastases resection (examples in Supplementary Figs. 9, 10). In order to accommodate the longitudinal evolution of consecutive DELFI-TF values in a single score, we calculated the DELFI-TF slope, which is defined as the slope of the fitted linear regression line to DELFI-TF values starting at the first liquid biopsy time point after treatment initiation and ending at the time of disease progression confirmed by RECIST1.1 (Supplementary Data 7). Patients with

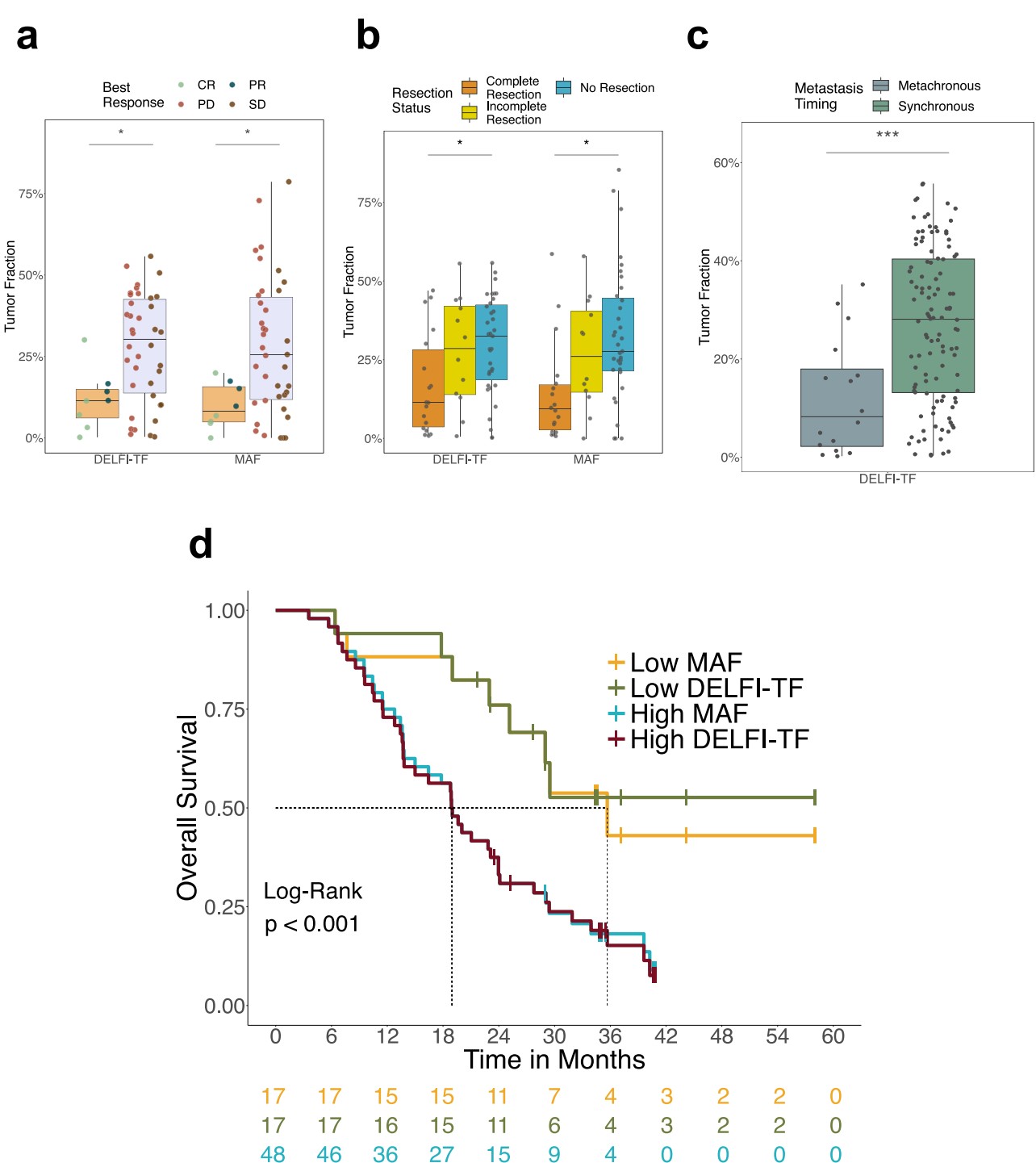

**Fig. 3 | DELFI-TF as a non-invasive biomarker for disease burden, systemic treatment response, and prognostic outcome. a** Tumor fractions assessed by DELFI-TF and MAF at baseline were significantly lower in patients with a later confirmed partial response (PR) or complete response (CR) by the first two consecutive RECIST1.1 measurements at follow-up (tan, $n = 8$), compared to cases with consecutive progressive disease (PD) or stable disease (SD) (lavender, $n = 38$) (DELFI-TF, two-sided Wilcoxon rank-sum, $p = 1.60$e-02; MAF, two-sided Wilcoxon rank-sum, $p = 2.2$e-02). **b** Tumor fractions assessed by DELFI-TF and MAF at baseline were significantly different among patients who were eventually treated with complete resection (orange; $n = 18$), incomplete resection (yellow; $n = 12$) or no resection (blue; $n = 35$) after receiving first-line systemic treatment (DELFI-TF, $p = 4.54$e-02, Kruskal–Wallis; MAF, $p = 1.09$e-02, Kruskal–Wallis). **c** Colorectal

cancer patients with metachronous metastases (gray; $n = 16$) exhibit lower tumor fractions assessed by DELFI-TF at baseline than patients who presented with synchronous metastases (green; $n = 112$) ($p = 5.08$e-04, two-sided Wilcoxon rank-sum). **d**, Kaplan–Meier curves for overall survival (OS) according to baseline DELFI-TF and MAF values. Patients with low DELFI-TF (green) or low MAF (yellow) experienced significantly longer OS than patients with high DELFI-TF (red) or high MAF (blue) ($p = 6.23$e-04, log-rank). Low DELFI-TF and MAF values were categorized as below the 25th percentile distribution of tumor fraction among the population of *RAS/BRAF* mutant individuals ($n = 65$). The middle hinge in the boxplots corresponds to the median, while the lower and upper hinges correspond to the first and third quartiles. The upper whisker extends from the hinge to the largest value no further than the 1.5 * interquartile range from the hinge.

**Table 1 | Multivariate Cox regression model for overall survival\***

| Description | Hazard ratio (95% Confidence interval) | p value |
|---|---|---|
| DELFI-TF | 9.84 (1.72–56.10) | 0.01 |
| Left v Right | 3.84 (2.15–6.83) | 0.0000049 |
| <65 v ≥65 | 1.37 (0.82–2.27) | 0.23 |
| CEA level | 1 (1.00–1.00) | 0.45 |
| SLD | 0.995 (0.99–1.00) | 0.22 |

\*Analyses performed for 125 individuals with available clinical or molecular data.

below-median DELFI-TF slopes had higher rates of objective radiologic responses (CR/PR) while on first-line treatment and longer durations of follow-up than patients with DELFI-TF slopes above the median (Fig. 4a).

Analysis of DELFI-TF slopes revealed that patients with below-median DELFI-TF slopes had longer PFS in the overall study population (13.5 months vs 10.5 months, HR = 2.32, 95% CI = 1.41–3.82, $p < 0.001$, log-rank) (Fig. 4b and Supplementary Data 8) and in patients who experienced durable clinical benefit, defined as an objective response or stable disease longer than 12 months (30.1 months vs 13.4 months, HR = 3.80, 95% CI = 1.67–8.68, $p < 0.001$, log-rank) (Supplementary Fig. 11). Clinical response evaluation using imaging could not distinguish between patients who ultimately were observed to have a partial response or stable disease (Supplementary Fig. 11). Patients with below-the-median DELFI-TF slopes experienced significantly longer overall survival (OS 62.8 months vs 29.1 months, HR = 3.12, 95% CI = 1.62-6.00, $p < 0.001$, log-rank) (Fig. 4c). Survival outcomes could be further stratified by DELFI-TF slopes and resection status. Patients with incomplete resection and DELFI-TF slope above the median experienced significantly shorter OS than patients with complete resection of the primary tumor and liver metastases and DELFI-TF slope below the median ($p < 0.001$, log-rank) (Supplementary Fig. 11).

## Discussion

In this study, we describe the development of DELFI-TF, a fragmentomics approach designed to measure tumor burden quantitatively, and demonstrate its potential for longitudinal disease monitoring in patients with cancer without the requirement of detecting mutations in tumor tissue. Despite diagnostic and treatment advances, most patients with metastatic disease have tumor progression[22], and there is an unmet need for a sensitive real-time assay to guide treatment. Currently, available follow-up methods, such as clinical imaging and serum proteins, have limited accuracy, and assessing treatment effectiveness after the start of therapy may be challenging[23]. Our results show that DELFI-TF has the potential to be more sensitive than conventional clinical approaches for monitoring treatment response, as DELFI-TF predicted PFS and OS better than serum CEA measurements and CT imaging both at baseline and after treatment initiation. Identifying treatment response or progression using DELFI-TF may provide opportunities to adapt a patient's treatment regimen and enhance patient outcomes.

DELFI-TF was similar in overall performance for non-invasive monitoring to a mutation-based MAF approach using known tissue-informed variants, but the ability to detect samples with independently validated ctDNA that was not detected by ddPCR suggests that DELFI-TF may be more accurate at capturing overall tumor burden. As tumor-informed approaches are logistically complex and may be confounded by tumor heterogeneity or clonal evolution during treatment[7,24,25], the tumor- and mutation-independent aspects of DELFI-TF provide distinct advantages. Additionally, since DELFI-TF does not require prior knowledge of somatic driver alterations, it could be applicable to samples from patients with other cancer types. Our

validation analyses in lung cancer, and a separate study highlights this capability in an immune checkpoint blockade trial in advanced cancer patients, including breast and other cancer types[26]. Although other WGS methods have been described for analyses of cfDNA[17,19,21], DELFI-TF more accurately represents tumor burden across the possible range of MAF levels.

The low-coverage WGS needed for fragmentation profiles, at the 6x coverage used in this study, is simpler to perform and less costly than targeted sequencing[27] and could be utilized more frequently during therapy than currently available approaches. The features used by DELFI-TF are assessed over large genomic regions, making them robust to overall sequencing depth, and other studies[26] have already shown that the approach can be performed at even lower (~2x) coverage. As the tumor burden can fluctuate over time, with lower levels after treatment response and increasing levels as the tumor develops resistance to therapy, comprehensive genomic profiling tests could be used to complement DELFI-TF to identify appropriate therapies at the time of progression.

In addition to the potential use of the DELFI-TF score as a predictor of outcome at a single time point, the current study highlighted the feasibility of tracking ctDNA dynamic changes throughout disease treatment. DELFI-TF slopes corresponding to aggregated measurements of consecutive DELFI-TF scores after treatment initiation were predictive of disease progression and death, even when adjusted by surgical outcomes. These findings support the potential application of DELFI-TF for real-time treatment monitoring and drug development assessment in prospective clinical trials. Additional research is needed to evaluate DELFI-TF over multiple time points to identify those which may provide the most useful surrogate endpoints of clinical response.

This study describes the development of DELFI-TF using a large cohort of *RAS/BRAF* mutant CRC patients with metastatic liver disease and the validation of the method in an independent cohort of CRC patients with *RAS/BRAF* wild-type tumors as well as in a cohort of lung cancer patients treated with immune checkpoint blockade. This study shows, using a well-organized phase III clinical trial, that tumor content as determined by cfDNA fragmentation is predictive of clinical outcome. Prospective studies are needed to determine the feasibility, efficacy and predictive value of monitoring longitudinal dynamics in both advanced cancers as well as for detection of minimal residual disease after surgery with curative intent. With these goals in mind, an observational prospective trial has been initiated to evaluate the efficacy of DELFI-TF for monitoring treatment response compared to radiologic imaging in patients with metastatic CRC (NCT02162563 for DOLPHIN). Overall, DELFI-TF provides a mutation and tumor-independent non-invasive approach for measuring tumor burden that may be useful for monitoring patients with colorectal and other cancers during therapy.

## Methods
### Study design and population
The phase III randomized CAIRO5 trial (NCT02162563)[24] investigates the optimal first-line systemic therapy for patients with histologically proven CRC with isolated, previously untreated, initially unresectable liver metastases. Patients treated with doublet chemotherapy (FOL-FOX or FOLFIRI) and bevacizumab with at least one blood draw prior to and after treatment initiation between March 2015 and November 2020 were included in the present study. All patients were considered unresectable at inclusion, *i.e.* R0-resection could not be achieved in one procedure with one surgical intervention. Upon treatment with doublet chemotherapy and bevacizumab, patients were evaluated every two months by an expert panel of liver surgeons and abdominal radiologists for the possibility of local treatment of colorectal liver metastases following current clinical practice. Clinical follow-up was performed according to the standard of care, including a clinical review every 3 months and CT imaging and serum CEA every six

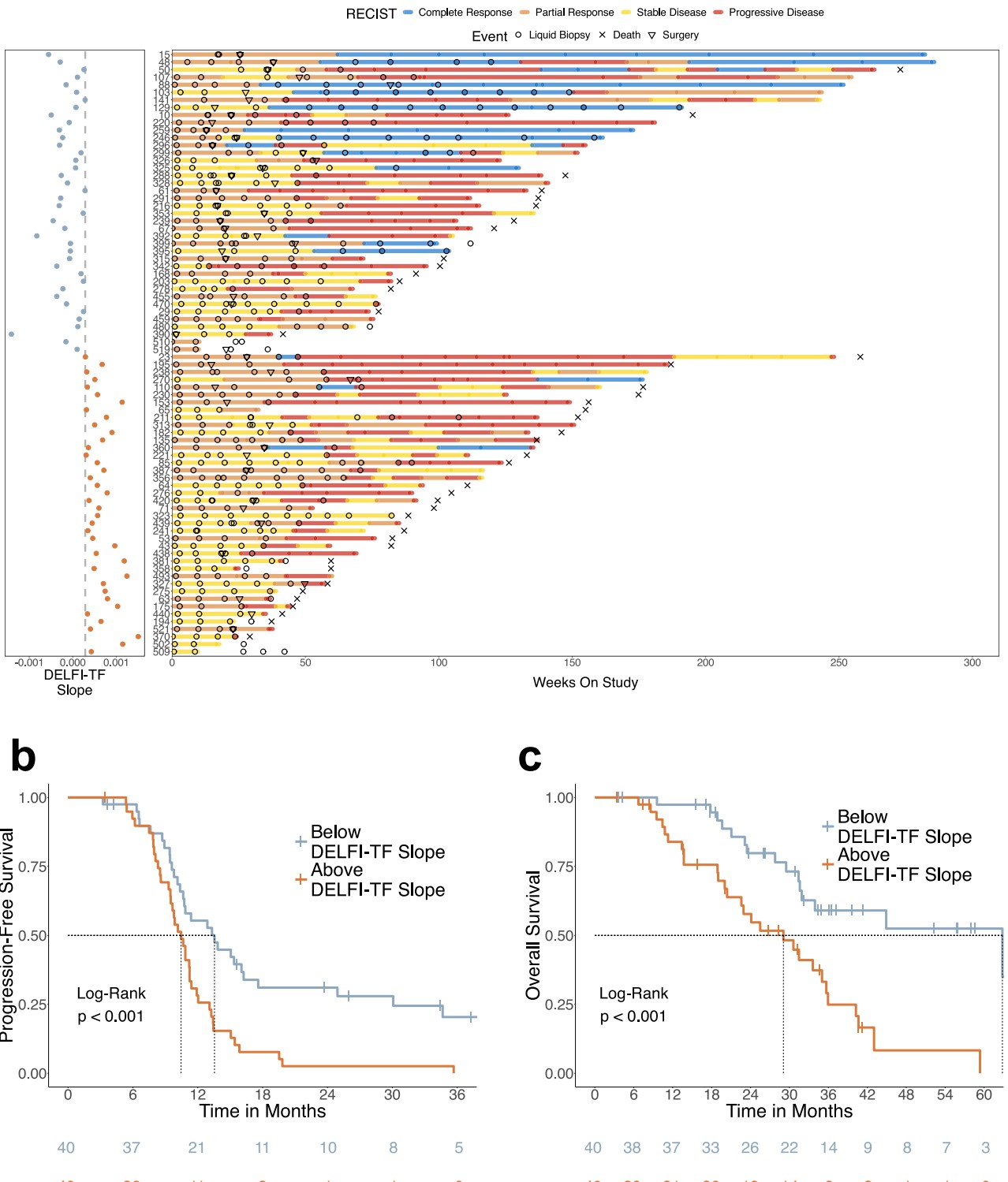

**Fig. 4 | Dynamic changes in DELFI-TF were associated with longitudinal clinical outcomes in colorectal cancer patients. a** DELFI-TF slopes were calculated amongst all patients with a post-treatment blood draw (T1) and a blood draw within 60 days of progression (Tn Progression:60). A linear regression model was applied using the difference in days between T1 and all time points up to progression ($n = 80$). Left, DELFI-TF slopes are colored based on results below (blue) or above (orange) the median value across all DELFI-TF slopes from eligible patients. Right, swimmer plot encompassing RECIST1.1, cfDNA testing events, surgery, and death events for patients according to time on the study since registration. Bar segments are colored according to the RECIST1.1 readouts. **b** Kaplan–Meier curves for progression-free survival (PFS) according to DELFI-TF below (blue) or above (orange) the median among patients with at least one blood draw within 60 days of disease progression ($n = 80$) ($p = 7.32$e-04, log-rank). **c**, Kaplan–Meier curves for OS according to DELFI-TF slopes below (blue) or above (orange) the median among patients with at least one blood draw within 60 days of disease progression ($n = 80$) ($p = 3.33$e-04, log-rank).

months. When the liver metastases stayed unresectable, chemotherapy was continued without the targeted agent for the total duration of pre- and post-operative treatment of six months, and patients were continuously evaluated until the progression of the disease by serum CEA and CT imaging every 2 months. Follow-up was recorded until September 1, 2021. The trial and follow-on studies were approved by a medical ethical committee of the Amsterdam University Medical Centre, performed according to the Declaration of Helsinki, and patients signed written informed consent for study participation and blood collection for translational research. Longitudinal plasma samples from lung cancer patients treated with chemotherapy and immune checkpoint blockade were obtained from Indivumed (Germany) with patient informed consent according to the Declaration of Helsinki.

## Blood collection and cfDNA extraction

Collection of liquid biopsy samples was performed at the medical center of inclusion prior to study treatment (baseline), pre-operatively, post-operatively, and every 3 months during follow-up until disease progression or treatment completion. Blood samples were obtained using 10 mL cell-free DNA BCT® tubes (Streck, La Vista, USA) and collected centrally at the Netherlands Cancer Institute (Amsterdam, the Netherlands) for CAIRO5 samples or at Indivumed (Hamburg, Germany) for samples from the lung cancer cohort. A two-step centrifugation process, 10 min at 1700×*g* and 10 min at 20,000×*g*, were used to isolate the cell-free plasma. The cell-free plasma was stored at −80 °C until further use. We aimed to use 4 mL of plasma for cfDNA isolation, which was feasible for the majority of the patients/time points. In case this was not feasible, we used a minimum of 1.5 mL. From the plasma, 60 uL of cfDNA was isolated and 9 uL in replicate (18 uL total) was used for the ddPCR assay, independent of the cfDNA concentration. cfDNA concentration was assessed using the double-strand DNA High-Sensitivity Qubit assay for all samples. The process for performing isolation of cfDNA from plasma was performed with the QIAsymphony robot, with an elution volume of 60 μL. Regardless of the cfDNA concentration, and per manufacturer's instructions, 9 μL of cfDNA sample was used as input into the ddPCR assay per duplicate.

## Library preparation and cfDNA sequencing

Aliquots of 15 ng cfDNA were used for the DELFI-TF analyses (requiring a minimum of 800 uL or 1 ng of cfDNA input) and ichorCNA analysis. NGS libraries were constructed using the NEBNext DNA Library Prep kit (New England Biolabs; Ipswich, MA, USA) with up to 15 ng of cfDNA input, as previously described[17], with modifications to the manufacturer's guidelines. AMPure XP beads (Beckman Coulter; Brea, CA, USA) were used exclusively for all library purification steps, in lieu of spin columns, and utilized an on-bead approach to minimize sample loss during elution and transfer steps. In this approach, AMPure XP beads were initially added during the end repair purification step, and the subsequent dA-tailing and adapter ligation reactions were conducted with beads present in the reaction mixture. Post-PCR purification was also performed using AMPure XP beads. cfDNA libraries were amplified using Phusion HotStart Polymerase (Thermo Fisher; Waltham, MA, USA). WGS library quality was determined using the 2100 Bioanalyzer (Agilent Technologies; Santa Clara, CA, USA) or the TapeStation 4200 (Agilent Technologies; Santa Clara, CA, USA). cfDNA libraries were pooled together and sequenced using a NovaSeq 6000 (Illumina; San Diego, CA, USA).

To limit batch effects, all time points collected from a single individual were processed together in a single library preparation batch to create genomic libraries, including a duplicate library as an inter-batch control and a technical replicate of nucleosomal DNA obtained from nuclease-digested human peripheral blood mononuclear cells as an intra-batch control (Supplementary Data 3).

Samples in the lung cancer validation set were processed separately from samples in the CAIRO5 study.

## RAS/BRAF mutation analyses

*RAS* and *BRAF V600* mutation analyses were performed on tumor tissue DNA following routine clinical practice. For the subset of patients with a *RAS/BRAF* tumor tissue mutation, longitudinal liquid biopsy hotspot mutation analyses by ddPCR (Bio-Rad, Hercules, CA, USA) and DELFI-TF fragmentation analyses were performed. The ddPCR™ KRAS G12/G13 (#1863506), ddPCR™ KRAS Q61 (#12001626), ddPCR™ KRAS A146T (#10049550), ddPCR™ BRAF V600 (#12001037), NRAS G12/G13 (#12001627), and NRAS Q61 (#12001006) Screening Kits were purchased from Bio-Rad and used according to the manufacturer's instruction, using 9 μL of sample, 11 μL of ddPCR supermix for probes (no dUTP), 1 μL of the multiplex assay and 1 μL of nuclease-free water. All measurements were performed in duplicate, including a blank (nuclease-free water) and a positive control. Patients with a RAS/BRAF mutation that could not be tracked by ddPCR because the variant identified on tumor tissue was not present on one of the available ddPCR Screening Kits were excluded (Supplementary Fig. 1). Data were analyzed using the QuantaSoft™ software version 1.6.6 (Bio-Rad, Hercules, CA, USA) and an automated correction algorithm as previously described in ref. 25. For the ddPCR assay, the limit of detection was determined based on the limit of blank, adjusting the outcome according to a predefined ratio of false-positive mutants found in WT samples, as described in a previous publication[28]. In all analyses, RAS/BRAF MT+ was defined as a ctDNA analysis with a positive result, that is, detectable mutant droplets above the limit of detection.

## Analyses of cfDNA sequencing data

On a per-sample basis, the paired-end sequenced reads were aligned to a reference genome (hg19) using paired-end alignment with Bowtie 2[29](version 2.4.2). Aligned reads were sorted, PCR duplicates were removed, and read pairs representing unique fragments were converted to BED format using Samtools[30] (version 1.13) and Bedtools[31](version 2.26.0), respectively. Fragment lengths were calculated based on start and end coordinates, and the fragments were divided into 504 5 Mb bins, covering ~2.6 Gb of the genome. We tiled the hg19 autosomes into 26,236 adjacent, non-overlapping 100-kb bins, excluding regions of low mappability and excluding reads that fell into publicly available blacklisted regions[17]. Using this approach, we excluded 361 Mb (13%) of the hg19 reference genome, including centromeric and telomeric regions. Short fragments were defined as having lengths between 100 and 150 bp, and long fragments as having lengths between 151 and 220 bp[17]. Next, the number of short and long fragments per bin was calculated using R/Bioconductor (version 3.6.2), and these counts were corrected by GC content[17]. The corrected count of short fragments was divided by the corrected count of long fragments by bin (short-to-long ratios) to obtain fragmentation profiles for each sample. We performed a principal component analysis (PCA) on the fragmentation profiles, retaining the top two principal components of variance between samples. Additional cfDNA-derived features included arm-level aneuploidy scores (a z-score calculated for each of 39 acrocentric arms[17], plasma-aneuploidy score (PA-score; 1 feature)[20], and the overall fragment-length distribution summarized by a 12-component mixture of normals (35 parameters). As previously described in ref. 20, the plasma-aneuploidy score (PA-score) was constructed from five chromosomes whose arms had the highest absolute z-scores and converted to *p* values (using a Student's *t* distribution with three degrees of freedom), and the negative of the sum of the logarithms of the *p* values was calculated for each sample.

Using the above features calculated for each sample, a random forest model was trained against the allele frequencies of the tumor-specific driver *RAS/BRAF* variant measured by ddPCR in the

longitudinal cfDNA samples. This model takes the mixture model weights (11 features), PA-score, the maximum absolute z-score scores (2 features), and the principal components of short-to-long ratios (2 features) as inputs and outputs a predicted MAF. In order to generate unbiased predictions, avoid overfitting, and assess generalizability, training was done via leave-one-patient-out cross-validation. In this cross-validation scheme, each patient's data was held out in turn, the model was trained on the remaining samples, and that trained model was then used to generate predictions for the held-out patient's samples. DELFI-TF was defined as the predicted MAF from this cross-validation scheme. We evaluated the quality of the generated predictions by assessing the correlation of these predictions with the observed ddPCR MAF values and by evaluating the relationship between those predictions and time to progression or death.

## DELFI-TF dynamics analysis

To capture the molecular dynamics of tumor burden over time, we computed the slope of the linear regression line fitted to the DELFI-TF values at time T1 and all subsequent time points until progression for the PFS analysis and up to 60 days after the progression date for the OS analysis. We limited the analysis to patients that had at least three samples before progression, and at least one sample collected in the progression window, which was the period 120 days before the progression date for PFS analysis (79 patients) and 120 days before and up to 60 days after the progression date for the OS analysis (80 patients). The regression lines were computed using Python/scikit-learn (version 3.9.13/1.1.1).

**IchorCNA methodology.** ichorCNA analysis code was obtained from GitHub (https://github.com/broadinstitute/ ichorCNA) and run using default parameters with the exception that the expectation-maximization algorithm was seeded to account for low tumor purity samples[21]. A panel of 30 plasma samples from healthy individuals was constructed for use as a panel of normals[17].

**Chromatin structure analysis.** A/B compartments were evaluated for colon adenocarcinoma (COAD) from TCGA and lymphoblastoid cells, as previously described in ref. 32. The median fragmentation profiles for ten CRC samples with high DELFI-TF values and ten randomly selected cancer-free individuals were constructed and compared across 100-kb bins. The median fragmentation profiles of the cancer-free individuals and the DELFI-TF of the individual CRC samples were used to extract an estimated median CRC component in the plasma. Copy-neutral regions were annotated if five or more of the ten CRC samples were copy-neutral in that region.

**Formalin-fixed paraffin-embedded (FFPE) tissue genomics.** Formalin-fixed paraffin-embedded (FFPE) tissue blocks from surgical resection or biopsies of the primary tumors were collected from all patients, and DNA was isolated from the available material using the Qiagen AllPrep DNA/RNA/miRNA Universal Kit (Qiagen, Hilden, Germany). Next, 250 ng of double-stranded genomic DNA was fragmented by Covaris shearing to obtain fragment sizes of 160–180 bp. Samples were purified using 2X Agencourt AMPure XP PCR Purification beads according to the manufacturer's instructions (Beckman Coulter; Brea, CA, USA; #A63881). The sheared DNA samples were quantified and qualified on a BioAnalyzer system using the DNA7500 assay kit (Agilent Technologies; Santa Clara, CA, USA; #5067-1506). With an input of a maximum of 1 μg sheared DNA, libraries for sequencing were constructed using the KAPA Hyper Prep Kit (KAPA Biosystems; Wilmington, MA, USA; #KK8504) and amplified using PCR. After library preparation, the libraries were purified using 1X AMPure XP beads, and the molarity was determined using a BioAnalyzer DNA7500 chip. All samples were sequenced on a NovaSeq S1 (Illumina; San Diego, CA, USA), single-read, 100 bp run, according to the manufacturer's

instructions. The resulting paired-end reads were aligned to hg19 with Bowtie 2[31] (version 2.4.5) and converted to BED format using Samtools[32] (version 1.6) and Bedtools[33] (version 2.30.0), respectively. Next, the fragment counts in non-overlapping 100-kb bins across the genome were GC-corrected and normalized using a panel of ten non-cancer samples and used for generating copy number profiles. Tumor copy number alterations were identified with Python (version 3.9.13).

**Lung cancer validation cohort targeted panel processing.** Longitudinal plasma samples collected from stage III and IV lung cancer patients were acquired from Indivumed (Indivumed Services GmbH; Hamburg, Germany). Plasma was collected in Streck BCT tubes. cfDNA was extracted from 3 to 4 mL of plasma using the MagMAX™ Cell-Free DNA Isolation kit (Life Technologies; Austin, TX, USA). Extracted cfDNA was examined using the TapeStation 4200 (Agilent Technologies; Santa Clara, CA, USA) and processed using the PGDx elio™ plasma complete kit (Personal Genome Diagnostics; Baltimore, MD, USA) following the manufacturer's protocol and recommendations for library preparation, hybridization, and sequencing. All longitudinal time points from a single patient were processed together on the same PGDx elio batch to avoid confounding batch with patient response. Raw sequencing data were analyzed through the PGDx elio server and automated pipeline.

**MaxMAF determination for lung cancer validation cohort.** We used the maximum mutant allele fraction (maxMAF) of somatic variants identified in the targeted panel analyses as a surrogate for ctDNA fraction in the lung cancer validation cohort. We first analyzed the PGDx elio plasma complete single nucleotide variant and insertion/deletion report to identify candidate mutations for calculating maxMAF. Candidate mutations were classified as somatic hotspots if the nucleotide changes were identical to an alteration observed in ≥20 cancer cases reported in the COSMIC database[24,33]. Putative germline mutations were identified as non-hotspots with MAF greater than 40%. As a stringent cutoff, we filtered all non-hotspot mutations and putative germline mutations from the PGDx elio plasma complete reports. Using the remaining somatic hotspot variants, we calculated the maxMAF. We then performed a correlation analysis between maxMAF and DELFI-TF for all samples in the lung cancer validation cohort.

## Statistical analyses

The Random Forest model was trained with scikit-learn (version 1.1.1) and Python (version 3.9.13). The percentages of variable contributions were determined by the Random Forest feature importance tool. Correlations between variables were calculated using Pearson's correlation coefficient. All hypothesis testing was performed using non-parametric tests (Wilcoxon rank-sum test, Kruskal–Wallis). Survival analyses were performed using Mantel-Cox log-rank tests. Analyses were performed with R Statistical Software (version 4.3.3 Foundation for Statistical Computing, Vienna, Austria). Unless otherwise noted, hypothesis tests were two-sided with a type 1 error of 5% for determining statistical significance. $P$ values in figures correspond to the following aliases: ($p < 0.0001 = ****, p < 0.001 = ***, p < 0.01 = **, p < 0.05 = *, p > 0.05 = $ ns).

## Reporting summary

Further information on research design is available in the Nature Portfolio Reporting Summary linked to this article.

# Data availability

The sequencing data generated in this study have been deposited in the database of the European Genome-Phenome Archive (EGA) and may be obtained at https://egaarchive.org/ under accession codes (EGAS00001006695 [https://ega-archive.org/studies/EGAS00001006695], EGAS00001005340). Requests for access

to the EGA deposited data should be addressed to the email dataaccesscommittee.pathology@nki.nl and will be evaluated by the NKI-AvL TGO data access committee and by the NKI-AvL IRB within 6 weeks following the request. Requests should meet GDPR requirements and are pending competitive research efforts. Positive evaluation is followed by establishing a data transfer agreement (~3 to 6 months). The processed data required to replicate the analyses herein are available in the supplementary data or GitHub repository.

## Code availability

Scripts for reproducing tables and figures in the manuscript are available in the GitHub repository https://github.com/delfidiagnostics/CAIRO5_Public.

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

## Acknowledgements

This work was supported in part by the Dr. Miriam and Sheldon G. Adelson Medical Research Foundation, Stand Up to Cancer-Dutch Cancer Society International Translational Cancer Research Dream Team Grant (SU2C-AACR-DT1415), Dutch Cancer Society grant 10438, the Gray Foundation, The Honorable Tina Brozman Foundation, the Commonwealth Foundation, SU2C in-Time Lung Cancer Interception Dream Team Grant, the Mark Foundation for Cancer Research, the Cole Foundation, a research grant from DELFI Diagnostics, and US National Institutes of Health grants CA121113, CA006973, CA233259, CA233259, CA062924, and CA271896. Stand Up To Cancer is a program of the Entertainment Industry Foundation administered by the American Association for Cancer Research. The funders had no role in study design, data collection and analysis, decision to publish, or preparation of the manuscript.

## Author contributions

R.J.A.F. and A.L. designed the initial experiment, and V.E.V. and R.J.A.F. supervised the study. I.V.E., B.A., K.L., Z.L.S., L.R., L.K.M., E.P., and T.W. performed analyses and interpreted data with input from J.C., S.C., T.W., E.P., K.B., C.J.A.P., J.T., P.B.B., N.C.D., G.A.M., R.B.S., V.E.V., R.J.A.F., and A.L. Samples and clinical information were provided by I.V.E., C.J.A.P., G.A.M., and R.J.A.F. Sequencing data were generated by B.C. The manuscript was written by A.L., I.V.E., Z.L.S., L.R., R.J.A.F., and V.E.V. with input from all authors who read and approved of the manuscript.

## Competing interests

A.L., S.C., R.B.S., and V.E.V. are inventors on patent applications submitted by Johns Hopkins University related to cell-free DNA for cancer detection. A.L., S.C., N.C.D., and R.B.S. are founders of DELFI Diagnostics, and R.B.S. is a consultant for this organization. V.E.V. is a founder of DELFI Diagnostics, serves on the Board of Directors, and owns DELFI Diagnostics stock, which is subject to certain restrictions under university policy. Additionally, Johns Hopkins University owns equity in DELFI Diagnostics. V.E.V. divested his equity in Personal Genome Diagnostics (PGDx) to LabCorp in February 2022. V.E.V. is an inventor on patent applications submitted by Johns Hopkins University related to cancer genomic analyses and cell-free DNA for cancer detection that have been licensed to one or more entities, including DELFI Diagnostics, LabCorp, Qiagen, Sysmex, Agios, Genzyme, Esoterix, Ventana, and ManaT Bio. Under the terms of these license agreements, the University and inventors are entitled to fees and royalty distributions. V.E.V. is an advisor to Viron Therapeutics and Epitope. These arrangements have been reviewed and approved by Johns Hopkins University in accordance with its conflict-of-interest policies. R.J.A.F. reports support from DELFI Diagnostics during the conduct of the study, as well as public- private partnership grants and other support from DELFI Diagnostics, Personal Genome Diagnostics, Cergentis BV, Natera and Merck BV outside the submitted work; in addition, R.J.A.F. has several patents pending. L.R., Z.L.S., B.A., J.C., K.L, L.K.M., C.P., T.W., E.P., and P.B.B., own DELFI Diagnostics stock. The remaining authors declare no competing interests.
