## [Peer Review File · Nature Communications]

Cancer treatment monitoring using cell-free DNA
fragmentomesEditorial Note: this manuscript has been previously reviewed at another journal that is not operating a transparent peer review scheme. This document only contains reviewer comments and rebuttal letters for versions considered at *Nature Communications*.

REVIEWERS' COMMENTS

Reviewer #3 (Remarks to the Author):

As discussed in my earlier review, the manuscript is well written, the bioinformatic approach is sound and the described quantitative assessment of tumor-derived DNA is of relevant clinical potential. The DNA fragmentation-based approach offers significant advantages in applicability and costs as compared to previously reported methodologies. All my concerns were addressed appropriately.

Reviewer #5 (Remarks to the Author):

In this study, the authors developed a novel approach, DELFI-TF, for predicting the cfDNA tumor fraction in the blood samples, the clinical value of the method was validated by independent studies. Overall, this is an interesting work, the authors have addressed most concerns of all Reviewers, I have no more questions.

Detailed Response to Reviewer Comments
Manuscript ID: NCOMMS-24-03968-T
Cancer treatment monitoring using cell-free DNA fragmentomes

Reviewers' Comments:

Reviewer #3 (Remarks to the Author):

As discussed in my earlier review, the manuscript is well written, the bioinformatic approach is sound and the described quantitative assessment of tumor-derived DNA is of relevant clinical potential. The DNA fragmentation-based approach offers significant advantages in applicability and costs as compared to previously reported methodologies. All my concerns were addressed appropriately.

Response: We thank the reviewer for their insights into this manuscript and for their positive feedback.

Reviewer #5 (Remarks to the Author):

In this study, the authors developed a novel approach, DELFI-TF, for predicting the cfDNA tumor fraction in the blood samples, the clinical value of the method was validated by independent studies. Overall, this is an interesting work, the authors have addressed most concerns of all Reviewers, I have no more questions.

Response: We thank the reviewer for their time and favorable response in reviewing this manuscript.